# Posthemorrhagic Hydrocephalus in Patients with Subarachnoid Hemorrhage Occurs Independently of CSF Osmolality

**DOI:** 10.3390/ijms241411476

**Published:** 2023-07-14

**Authors:** Sara Diana Lolansen, Nina Rostgaard, Tenna Capion, Nicolas H. Norager, Markus Harboe Olsen, Marianne Juhler, Tiit Illimar Mathiesen, Nanna MacAulay

**Affiliations:** 1Department of Neuroscience, University of Copenhagen, DK-2200 Copenhagen, Denmark; 2Department of Neurosurgery, University Hospital of Copenhagen—Rigshospitalet, DK-2100 Copenhagen, Denmark; 3Department of Neuroanaesthesiology, University Hospital of Copenhagen—Rigshospitalet, DK-2100 Copenhagen, Denmark; 4Department of Clinical Medicine, University of Copenhagen, DK-2100 Copenhagen, Denmark; 5Department of Clinical Neuroscience, Karolinska Institutet, 17177 Stockholm, Sweden

**Keywords:** cerebrospinal fluid, osmolality, subarachnoid hemorrhage, posthemorrhagic hydrocephalus

## Abstract

The molecular mechanisms underlying the development of posthemorrhagic hydrocephalus (PHH) remain incompletely understood. As the disease pathogenesis often cannot be attributed to visible cerebrospinal fluid (CSF) drainage obstructions, we here aimed to elucidate whether elevated CSF osmolality following subarachnoid hemorrhage (SAH) could potentiate the formation of ventricular fluid, and thereby contribute to the pathological CSF accumulation observed in PHH. The CSF osmolality was determined in 32 patients with acute SAH after external ventricular drainage (EVD) placement and again upon EVD removal and compared with the CSF osmolality from 14 healthy control subjects undergoing vascular clipping of an unruptured aneurism. However, we found no evidence of elevated CSF osmolality or electrolyte concentration in patients with SAH when compared to that of healthy control subjects. We detected no difference in CSF osmolality and electrolyte content in patients with successful EVD weaning versus those that were shunted due to PHH. Taken together, elevated CSF osmolality does not appear to underlie the development of PHH following SAH. The pathological CSF accumulation observed in this patient group must thus instead be attributed to other pathological alterations associated with the abnormal presence of blood within the CSF compartments following SAH.

## 1. Introduction 

Cerebrospinal fluid (CSF) continuously circulates through the central ventricular system and subarachnoid space of the mammalian brain. Disturbances in CSF homeostasis causing an imbalance between the production and reabsorption of CSF may promote pathological CSF accumulation, dilation of the cerebral ventricles, and the clinical entity hydrocephalus [1]. The pathological CSF accumulation may dangerously elevate the intracranial pressure (ICP) and compress the surrounding brain parenchyma. If left untreated, hydrocephalus may lead to neurological compromise or coma, and ultimately fatal brain herniation [2]. Although lifesaving neurosurgical procedures designed to divert the excess CSF are available, complications, failures, and substantial patient morbidity frequently follow such procedures [3,4]. Sufficiently efficient pharmacological treatments remain unavailable, although fibrinolytic therapy may be applicable in the resolution of hematomas and alleviation of possible CSF drainage obstructions [5,6,7]. The current lack of routinely used clinical pharmacotherapies derives, at least in part, from our incomplete understanding of the disease pathogenesis. Although some forms of hydrocephalus arise from mechanical obstructions causing diminished CSF reabsorption, hydrocephalus often manifests in the absence of discernible obstructions [8,9]. Posthemorrhagic hydrocephalus (PHH), one of the most common forms of hydrocephalus, can develop at any age as a complication of brain hemorrhage [10] and commonly presents in the absence of discernible CSF outflow obstructions on diagnostic imaging [9,11,12]. PHH often occurs in relation to hemorrhagic stroke, e.g., subarachnoid hemorrhage (SAH), or traumatic brain injury [13]. Amongst a complex wealth of pathological alterations, hemorrhage within the ventricular cavities causes ventricular wall damage, disruption of the underlying ventricular/subventricular zone, and inflammatory alterations in the CSF-secreting choroid plexus in the ventricular system [14,15,16,17,18,19,20]. The inflammatory choroid plexus alterations promote CSF hypersecretion and acute hydrocephalus in animals [16,17,18,21] and are speculated to underlie the acute CSF accumulation observed in patients with PHH. However, the molecular mechanisms driving the pathological CSF accumulation remain incompletely understood. 

CSF osmolality disturbances have long been speculated to play a role in the development of hydrocephalus [22]. Although the CSF osmolality resembles that of plasma in healthy animals and humans [23], experimentally elevated CSF osmolality can increase the rate of CSF secretion [23,24,25] and is sufficient to induce hydrocephalus in experimental animals [26,27]. Elevated CSF osmolality could thus potentially contribute, at least in part, to the pathological CSF accumulation observed in PHH and could arise with osmotically active substances originating from the hemorrhagic event. CSF from animals and patients with PHH contains elevated levels of inflammatory markers [17,18,20,28], blood breakdown products [29,30,31,32], and various extracellular matrix components and cell adhesion molecules [32,33], which upon degradation could elevate the CSF osmolality. Taken together, such CSF osmolality elevation could be sufficient to contribute to the pathological CSF accumulation observed in PHH. However, at present, whether the CSF osmolality is elevated in PHH remains an area of controversy [34,35,36] and has occasionally been extrapolated based on analyses using lumbar CSF or a mix of lumbar and ventricular CSF [35,36], the composition of which may differ from ventricular CSF [37,38,39,40], or with a lack of a control group [34]. 

CSF osmolality could, if associated with PHH, also serve as a potential biomarker of PHH severity. At present, many patients with PHH undergo shunt insertion due to the development of chronic hydrocephalus after the acute phase. A need to shunt patients is determined by clinical assessment and unsuccessful weaning from external ventricular drains (EVDs) [41]. In this context, elevated CSF osmolality is a potential marker, which could serve to shorten the duration of temporary management procedures such as EVDs in patients that eventually require shunt insertion due to chronic PHH progression, thereby minimizing the risk of infection and potential adverse effects [42,43,44,45]. 

We here tested the hypotheses that CSF osmolality is associated with PHH or with chronic hydrocephalus after acute PHH. We assessed the CSF osmolality in patients with aneurysmal SAH, one of the most common causes of PHH, by comparison to CSF collected from healthy control subjects and determined whether CSF osmolality was associated with the development of chronic hydrocephalus requiring permanent CSF diversion. 

## 2. Results 

### 2.1. Patients with SAH Demonstrate Unaltered CSF Osmolality 

To determine whether the pathological CSF accumulation observed in patients with SAH could arise from an elevated CSF osmolality, we compared the osmolality of ventricular CSF samples collected upon EVD placement in patients with SAH (start samples) to that obtained from the cisternal compartment in healthy control subjects undergoing vascular clipping for an un-ruptured aneurysm. The CSF osmolality in patients with SAH (292 ± 10 mOsm, *n* = 29) was not significantly different from that obtained in healthy control subjects (290 ± 7 mOsm, *n* = 14, *p* = 0.44, Figure 1a). Accordingly, similar CSF concentrations of Na^+^, one of the main CSF osmolytes, were found in the SAH patient group (148 ± 14 mM, *n* = 32) and the group of healthy control subjects (147 ± 12 mM, *n* = 14, *p* = 0.40, Figure 1b). CSF concentrations of Ca^2+^ were likewise similar (Appendix A). Taken together, we found no differences in CSF osmolality between patients with SAH and the healthy control subjects. 

### 2.2. Patients with SAH Demonstrate CSF Glucose and Lactate Disturbances 

Despite the unaltered CSF osmolality, patients with SAH could still exhibit CSF disturbances originating from the sudden entry of blood-derived metabolites and other substances into the CSF compartments during the hemorrhagic event. To quantify CSF disturbances as previously observed by others [35,36,46], CSF glucose and lactate levels were determined. These analyses revealed elevated levels of glucose (SAH: 4.14 ± 0.77 mM, *n* = 32; ctrl: 3.24 ± 0.86 mM, *n* = 14, *p* < 0.001, Figure 1c) and lactate (SAH: 2.53 ± 1.39 mM, *n* = 32; ctrl: 0.97 ± 0.25 mM, *n* = 14, *p* < 0.001, Figure 1d) in CSF from patients with SAH compared to that obtained from healthy control subjects. No correlation was found between CSF osmolality, Na^+^, glucose, and lactate levels and patient age or time from ictus to CSF sampling (Appendix A).

### 2.3. CSF Disturbances in Patients with SAH Do Not Predict Development of Chronic Hydrocephalus

To determine whether shunted SAH patients exhibited more pronounced CSF osmolality disturbances in comparison to SAH patients successfully weaned off from EVD, we compared the CSF osmolality of the two patient groups using the CSF samples collected upon hospitalization (start samples). However, the CSF osmolality in patients with SAH who underwent shunt insertion due to chronic PHH development (289 ± 10 mOsm, *n* = 10) did not differ from that obtained from patients with successful EVD weaning (293 ± 9 mOsm, *n* = 19, *p* = 0.32, Figure 2a). Accordingly, the CSF concentration of Na^+^ did not differ significantly between the two patient groups (weaned: 148 ± 15 mM, *n* = 20; shunt: 149 ± 12 mM, *n* = 12, *p* = 0.26, Figure 2b) and the CSF concentration of Ca^2+^ was likewise similar (Appendix A). No additional differences in CSF content were detected in the two SAH patient groups, as reflected by the similar levels of glucose (weaned: 4.23 ± 0.74 mM, *n* = 20; shunt: 3.99 ± 0.83 mM, *n* = 12, *p* = 0.41, Figure 2c) and lactate (weaned: 2.21 ± 0.68 mM, *n* = 20; shunt: 3.06 ± 2.03 mM, *n* = 12, *p* = 0.40, Figure 2d). Taken together, CSF osmolality, Na^+^, Ca^2+^, glucose, and lactate content were similar in patients who developed chronic PHH following SAH and those who did not. 

### 2.4. CSF Osmolality Does Not Reflect SAH Severity or Neurological Status 

To determine whether CSF osmolality disturbances were more pronounced in patients with severe SAH and reflected the patient’s level of consciousness, CSF osmolality as a function of the modified Fisher’s scale and the World Federation of Neurological Surgeons (WFNS) grading scale was assessed using the CSF samples collected upon hospitalization (start samples) from the entire SAH group (weaned and shunted). The CSF osmolality did not differ across the modified Fisher’s scale (Grade 1: 282 mOsm, *n* = 2; Grade 2: 297 ± 14 mOsm, *n* = 4; Grade 3: 296 ± 4 mOsm, *n* = 7; Grade 4: 290 ± 9 mOsm, *n* = 16, *p* = 0.47, Figure 3a) or the WFNS grading scale (Grade 1: 290 ± 8 mOsm, *n* = 9; Grade 2: 292 ± 13 mOsm, *n* = 8; Grade 3: NA; Grade 4: 288 ± 11 mOsm, *n* = 5, Grade 5: 297 ± 4 mOsm, *n* = 7, *p* = 0.27, Figure 3b). Taken together, CSF osmolality was similar across the groups, regardless of SAH severity or the patient’s neurological status upon hospitalization. 

### 2.5. CSF Glucose and Lactate Disturbances, but Not Osmolality, Decline over Time 

To elucidate the dynamics of the observed CSF alterations in patients with SAH and determine whether these CSF alterations differed between the weaned and shunted patients, paired CSF samples collected upon hospitalization (start samples) and just prior to EVD removal (end samples) were compared by assessing CSF osmolality, Na^+^, Ca^2+^, glucose, and lactate content. Across the whole SAH patient group, separation of CSF osmolality measurements into start and end samples revealed no significant difference in CSF osmolality (start: 292 ± 9 mOsm; end: 292 ± 13 mOsm, *n* = 28 of each, *p* = 0.28, Figure 4a, left panel). Determination of the mOsm change for each individual SAH patient revealed no difference from the start sample to end sample (0.1 ± 18 mOsm, *n* = 28, *p* = 0.28, Figure 4a, right panel). Accordingly, no significant difference in CSF Na^+^ concentration was detected upon separation into the start and end samples (start: 149 ± 14 mM; end: 147 ± 15 mM, *n* = 31 of each, *p* = 0.65, Figure 4b, left panel) with no significant change within each individual patient between the two samples (2 ± 22 mM Na^+^, *n* = 31, *p* = 0.65, Figure 4b, right panel). The CSF concentration of Ca^2+^ was likewise similar in the start and end samples (Appendix A). Quantification of CSF glucose and lactate changes revealed significantly lower glucose levels in the end samples (3.03 ± 0.72 mM vs. 4.14 ± 0.78 mM in the start samples, *n* = 31 of each, *p* < 0.001, Figure 4c, left panel), amounting to an average decrease of 1.11 ± 1.00 mM glucose (*n* = 31, *p* < 0.001, Figure 4c, right panel). CSF lactate levels were also significantly lower in the end samples (1.79 ± 0.52 mM vs. 2.54 ± 1.41 mM in the start samples, *n* = 31 in each, *p* < 0.001, Figure 4d, left panel) with an average decrease of 0.75 ± 1.28 mM lactate (*n* = 31, *p* < 0.001, Figure 4d, right panel). Separation of the SAH patient group into weaned and shunted SAH patients yielded similar observations (Appendix A). The osmolality was stable over time in SAH patient CSF, while glucose and lactate decreased similarly in patients regardless of successful weaning or shunting. No correlation was found between end sample CSF osmolality, Na^+^, and lactate levels and the time between the start and the end CSF samples. However, a weak correlation for glucose was apparent (Appendix A), substantiating that CSF glucose levels decline over time. 

## 3. Discussion 

We found no evidence of elevated CSF osmolality in patients with SAH when compared to healthy control subjects and no detectable CSF osmolality difference between shunted SAH patients and patients with successful EVD weaning. We thus conclude that PHH following SAH appeared un-associated with elevated CSF osmolality, promoting water flow into the CSF compartments, and thereby does not contribute to the development of SAH-related hydrocephalus.

The CSF osmolality resembles that of plasma in healthy animals and humans [23]. Normal CSF secretion consequently appears not to rely on osmotically obliged water movement [23] but rather transporter-mediated water transport [23,47,48]. Although experimentally elevated CSF osmolality is sufficient to induce hydrocephalus in experimental animals [26,27], such experimental induction relies on either acute delivery of up to ten-fold higher bolus administration of osmotic challenges than that of healthy animals or, alternatively, continuous ventricular infusion of CSF with elevated osmolality [26,27]. We here observed undisturbed osmolality in CSF samples obtained during the placement of EVD in patients with SAH compared to CSF obtained from the basal cisterns in healthy subjects undergoing preventive clipping of an unruptured aneurysm. The differential sampling site is ethically dictated and represents a limitation to the study, but whether the CSF content of the two compartments is compatible remains unclear, as CSF comparison studies remain largely limited to ventricular versus lumbar CSF [37,49], which, however, are of similar osmolality [49]. The CSF Na^+^ (and Ca^2+^) concentration was, likewise, similar in the two SAH patient groups, suggesting no overall electrolyte disturbance, which aligns with the gross similarity of the ionic content of blood entering during the hemorrhagic event and the resident CSF. The CSF osmolality is dominated by the electrolyte contribution, as illustrated by the nearly 150 mM Na^+^ detected in the CSF samples. According to the principle of electroneutrality, this cationic contribution must be matched by an accompanying anion; in the CSF, mainly Cl^−^ and HCO_3_^−^, when taken together approach the ~290 mOsm observed in human CSF (this study and others [23,36]). It has, therefore, in the context of PHH, been proposed that a hemorrhage-associated elevation of CSF osmolality would arise with a potential breakdown of the blood protein entering the CSF compartments. Although the blood protein content is substantial when determined as a mass, its osmotic contribution is relatively minor due to the large molecular weights of the main blood protein constituents: blood consists of approximately 55% plasma and 45% red blood cells [50], with albumin (66 kDa) being the main protein in plasma [50], around 6–8 g/100 mL blood [51], and hemoglobin (65 kDa) as the main (95%) protein in red blood cells, around 15 g/100 mL blood [52]. With these numbers, one obtains the protein contribution to the blood osmolality of around 1.5 mOsm, which is dwarfed by the osmolality of human plasma around 290 mOsm [23], mainly assigned to the blood electrolytes. Considering an estimated entry of approximately 15 mL of blood into the ventricular system in SAH [53], which is diluted in the 150 mL CSF space in the adult human [54], the hemorrhage-induced protein-based contribution to CSF osmolality amounts to 0.15 mOsm. Even with an assumed breakdown of blood protein in CSF, which will be gradually drained from the CSF with its flow through the ventricular system, the contribution to the overall osmotic load of the CSF is predicted to be minor. With the limited osmotic influence on the rate of CSF secretion in experimental rats (where the CSF secretion rate increased with a mere 0.4% per mOsm increase in the ventricular space) [23], we suspect that a hemorrhage-mediated protein-based contribution to CSF osmolality does not suffice to generate PHH in patients with SAH. These considerations align well with the present observations of comparable CSF osmolality in patients with SAH and healthy control subjects and the absence of any detectable CSF osmolality difference between shunted SAH patients and patients with successful EVD weaning. However, two previous studies reported elevated CSF osmolality in adult patients with SAH [36] and preterm infants with PHH following intraventricular hemorrhage (IVH) [35]. Both studies demonstrated concurrent elevation of various CSF electrolytes [35,36], which was not observed in the present CSF samples. These differences may possibly be explained by the use of lumbar CSF samples [35] or comparison of ventricular patient CSF to lumbar control CSF [36], the composition of which may differ to that of ventricular/cisternal CSF, which was analyzed in the present study. Noticeably, a recent study revealed an extraordinarily low CSF osmolality in the majority of tested adult patients with IVH, frequently more than 100 mOsm below the 290 mOsm usually reported for human CSF [34]. However, this study employed a remarkably small CSF volume (10 µL) for their osmolality determinations, which may introduce technical errors, and did not employ control CSF [34]. 

At present, it remains unclear why some patients with SAH can be successfully weaned from their EVDs, while a considerable patient cohort fail EVD weaning and need to undergo shunt insertion due to chronic PHH development. Identification of prognostic biomarkers for the development of PHH is thus greatly needed. However, as the present study revealed similar CSF osmolality in shunted SAH patients and patients with successful EVD weaning, CSF osmolality cannot be employed as a prognostic biomarker of PHH development, nor explain why some patients require a shunt and others not. Moreover, CSF osmolality did not reflect the severity of the SAH nor the patient’s neurological status upon hospitalization. It must be noted that division of the patients into the weaned and the shunted groups was based on clinical management; patients who failed 2–3 attempts of weaning or closure of an EVD underwent shunting. Another limitation lies in the retrospective nature of the study and the small study groups that were not matched for age or sex, which can induce variability and subsequent failure to detect small differences between groups. However, with the low osmotic water permeability of the rodent choroid plexus epithelium [23], we predict that such putative small differences would be insufficient in driving pathologically excessive fluid flow across this tissue. Another study limitation resides in the technically determined selective quantification of a select subset of electrolytes (Na^+^ and Ca^2+^). We, therefore, cannot exclude that the concentration of other prominent CSF electrolytes, such as Cl^−^ or HCO_3_^−^, may have been altered. However, such an anionic increase would require a matching cation contribution, which, if substantial, should be reflected in the CSF osmolality. 

The present findings of 1–2 mM elevation in CSF glucose and lactate in patients with SAH may derive from the extravasation of blood into the CSF compartments. Presence of living red blood cells within the CSF could, in addition, elevate the lactate levels as a result of their anaerobic metabolism [55]. Alternatively, cerebrovascular events frequently cause local cerebral hypoperfusion and altered brain metabolism, which may promote the release of metabolites such as lactate [56,57]. Elevated CSF lactate is a common feature of SAH, especially in patients with worse clinical outcomes [36,58,59], and could as such serve as a future potential biomarker for clinical outcomes. The declining CSF lactate levels observed after SAH may reflect gradual normalization of energy metabolism, rather than dilution by CSF flow. The elevated CSF glucose and lactate levels in patients with SAH corroborate earlier findings in adult patients with SAH [36], but contradict other findings of decreased CSF glucose levels in preterm infants with IVH [35,46]. Although the reason for this apparent discrepancy remains unclear, CSF disturbances may potentially differ according to the etiology of the hemorrhage (SAH [36] versus IVH [35,46]), the age at which the hemorrhagic event occurs (prematurity [35,46] versus adulthood [36]), and the CSF sample site (ventricular/cisternal (this study) versus lumbar [35,46]), the composition of which may differ [37,38,39,40]. 

In conclusion, elevated CSF osmolality does not appear to underlie the development of PHH following SAH. The pathological CSF accumulation observed in this patient group may thus instead be attributed to other pathological alterations associated with the abnormal presence of blood within the CSF compartments, such as choroid plexus CSF hypersecretion or impaired CSF drainage, possibly driven by post-hemorrhagic inflammation [9,11,16,18]. 

## 4. Materials and Methods 

### 4.1. Patients and CSF Sample Collection 

Patients with SAH who were treated with EVD between June 2019 and September 2021 at the Department of Neurosurgery, The Neuroscience Centre, Copenhagen University Hospital, Rigshospitalet, Copenhagen, Denmark, were retrospectively included in the present study, which employed CSF samples stored as part of a prospective trial on EVD weaning (clinical trial identifier: NCT03948256 [60]). Only adult patients with CSF samples obtained at the time of placement of the EVD (see below) and upon termination of the EVD, with no signs of infection in that interval, were included in the study. Upon hospitalization, the patient’s level of consciousness was assessed by the WFNS grading scale (Grade 1: *n* = 10; Grade 2: *n* = 8; Grade 3: *n* = 0; Grade 4: *n* = 7; Grade 5: *n* = 7) [61] and the radiographic appearance of the SAH scored by the modified Fisher’s scale (Grade 1: *n* = 2; Grade 2: *n* = 4; Grade 3: *n* = 7; Grade 4: *n* = 19) [62]. All patients had acute hydrocephalus, as clinically evaluated with computed tomography scanning. 

CSF samples were collected from a total of 32 patients with SAH (mean age: 60 y; range: 27–77 y; F/M: 20/12) through their EVD into a sterile collection tube. The first CSF sample (‘start sample’, around 5 mL CSF) was obtained in the acute phase after placement of the EVD, either within 24 h of ictus (*n* = 21) or as soon as possible hereafter as a sample taken directly from the EVD system (*n* = 11, mean: 44 h; range: 25–74 h). The last CSF sample (‘end sample’, 1–2 mL CSF) was obtained either just before the removal of the EVD, or during the placement of a ventriculoperitoneal shunt. The interval between the start and end samples was on average 19 days, with a range of 5–45 days. After collection, the CSF samples were immediately centrifuged at 2000× *g* for 10 min the supernatant aliquoted in polypropylene microtubes (Sarstedt, Nümbrecht, Germany) before storage at −80 °C [63]. 

Of the 32 patients with SAH, 20 patients were successfully weaned off the EVD (‘weaned’), while the remaining 12 patients underwent ventriculoperitoneal shunt surgery due to chronic PHH (‘shunt’). The control group was 14 patients with unruptured aneurysms undergoing preventive surgery (vascular clipping) (mean age: 61 y, range: 39–71 y, 8F/6M), from whom CSF was collected from the basal cisterns during surgery prior to clipping of the aneurysm.

The study was approved by the Ethics committee of the Capital Region of Denmark (H-19001474, 14 March 2019 and H-17011472/69197, 22 March 2019) and the Danish Data Protection Agency (VD-2019-210, 8 April 2019). Oral and written informed consent were obtained from all patients or their next of kin depending on the capacity of the patients. Aliquots of the CSF samples have been analyzed for other components in unrelated studies [18,64,65]. 

### 4.2. CSF Osmolality, Electrolyte, and Metabolite Measurements

To prevent breakdown of CSF solutes, the CSF samples were stored at −80 °C upon collection and care was taken to only thaw the CSF samples again right before the measurements. For the CSF osmolality measurements, 100 µL of each CSF sample was transferred to osmometer Eppendorf tubes (Sarstedt, Nümbrecht, Germany) and the CSF osmolality determined with a freezing point depression osmometer (Löser, Type 15, Berlin, Germany) with an accuracy of ±1 mOsm. As soon as the CSF samples were thawed and aliquoted for osmolality determination, the remaining CSF was immediately put back on ice and CSF Na^+^, Ca^2+^, glucose, and lactate levels were quantified with an ABL90 FLEX blood gas analyzer (Radiometer, Copenhagen, Denmark).

### 4.3. Statistics

Data analysis and statistical tests were conducted in GraphPad Prism (GraphPad Software, v 9). Data were tested for normality with the Shapiro–Wilk test. For unpaired data, an unpaired two-tailed *t*-test or a Mann–Whitney test was conducted. For paired data, a paired two-tailed *t*-test or a Wilcoxon matched-pairs signed rank test was conducted. For determination of average CSF changes (osmolality, Na^+^, Ca^2+^, glucose, and lactate), a one-sample two-tailed *t*-test or a one-sample Wilcoxon signed rank test was conducted. For more than two groups, a Kruskal–Wallis test was conducted followed by Dunn’s multiple comparisons test. Statistical tests are indicated in the figure legend. *p* < 0.05 was considered statistically significant. All data are presented as mean ± standard deviation, and *n* corresponds to the number of individual patients (notice that *n* varies as CSF osmolality could not be quantified in four CSF samples (three start samples and one end sample), while CSF Na^+^, Ca^2+^, glucose, and lactate levels could not be quantified in one CSF end sample) due to insufficient sample volumes. 

## Figures and Tables

**Figure 1 ijms-24-11476-f001:**
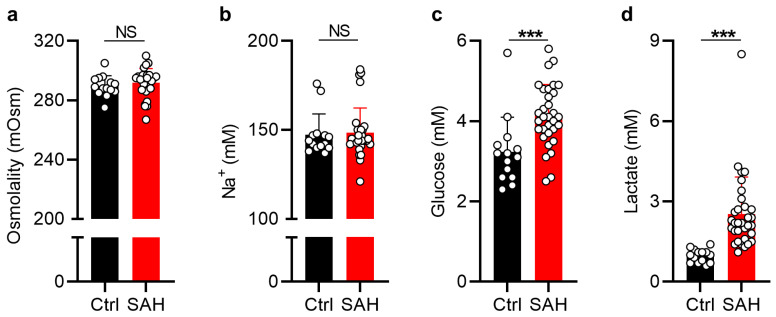
**Patients with SAH demonstrate unaltered CSF osmolality and electrolyte content but elevated glucose and lactate.** (**a**) CSF osmolality in patients with SAH (*n* = 29) and healthy control subjects (*n* = 14). (**b**) CSF Na^+^ concentration in patients with SAH (*n* = 32) and healthy control subjects (*n* = 14). (**c**) CSF glucose levels in patients with SAH (*n* = 32) and healthy control subjects (*n* = 14). (**d**) CSF lactate levels in patients with SAH (*n* = 32) and healthy control subjects (*n* = 14). Data are presented as mean ± standard deviation and statistical significance was tested with an unpaired two-tailed *t*-test or a Mann–Whitney test, depending on normality. Ctrl; control, SAH; subarachnoid hemorrhage. *** *p* < 0.001. NS = not significant.

**Figure 2 ijms-24-11476-f002:**
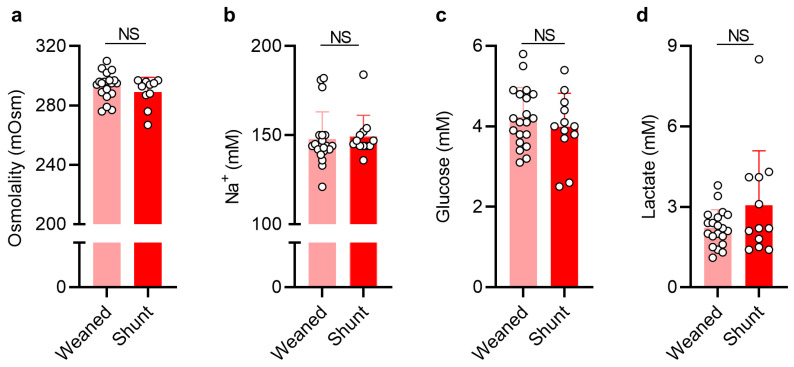
**CSF disturbances do not reflect development of PHH**. (**a**) CSF osmolality in weaned (*n* = 19) and shunted SAH patients (*n* = 10). (**b**) CSF Na^+^ concentration in weaned (*n* = 20) and shunted SAH patients (*n* = 12). (**c**) CSF glucose levels in weaned (*n* = 20) and shunted SAH patients (*n* = 12). (**d**) CSF lactate levels in weaned (*n* = 20) and shunted SAH patients (*n* = 12). Data are presented as mean ± standard deviation and statistical significance was tested with an unpaired two-tailed *t*-test or a Mann–Whitney test, depending on normality. NS = not significant.

**Figure 3 ijms-24-11476-f003:**
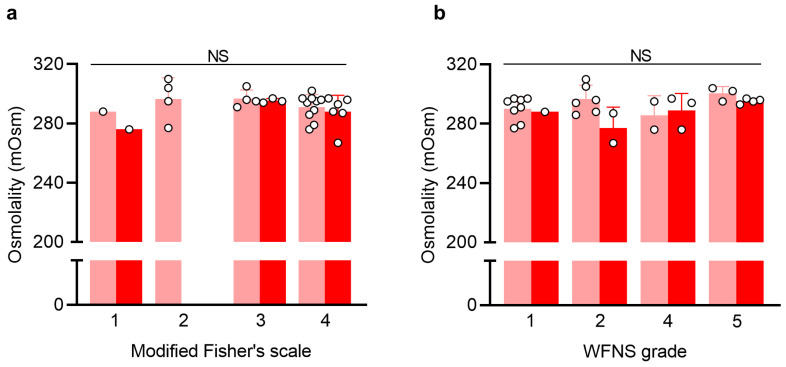
**CSF disturbances do not reflect SAH severity or the patient’s neurological status**. (**a**) CSF osmolality as a function of the modified Fisher’s scale in weaned (*n* = 19) and shunted SAH patients (*n* = 10). (**b**) CSF osmolality as a function of WFNS grade in weaned (*n* = 19) and shunted SAH patients (*n* = 10). Data are presented as mean ± standard deviation and statistical significance was tested with a Kruskal–Wallis test followed by Dunn’s multiple comparisons test (*p* ≥ 0.05 for all multiple comparison tests conducted between weaned and shunted SAH patients, not indicated on panels a and b for clarity). Light red; weaned patients, red; shunted patients, WFNS; World Federation of Neurological Surgeons. NS = not significant.

**Figure 4 ijms-24-11476-f004:**
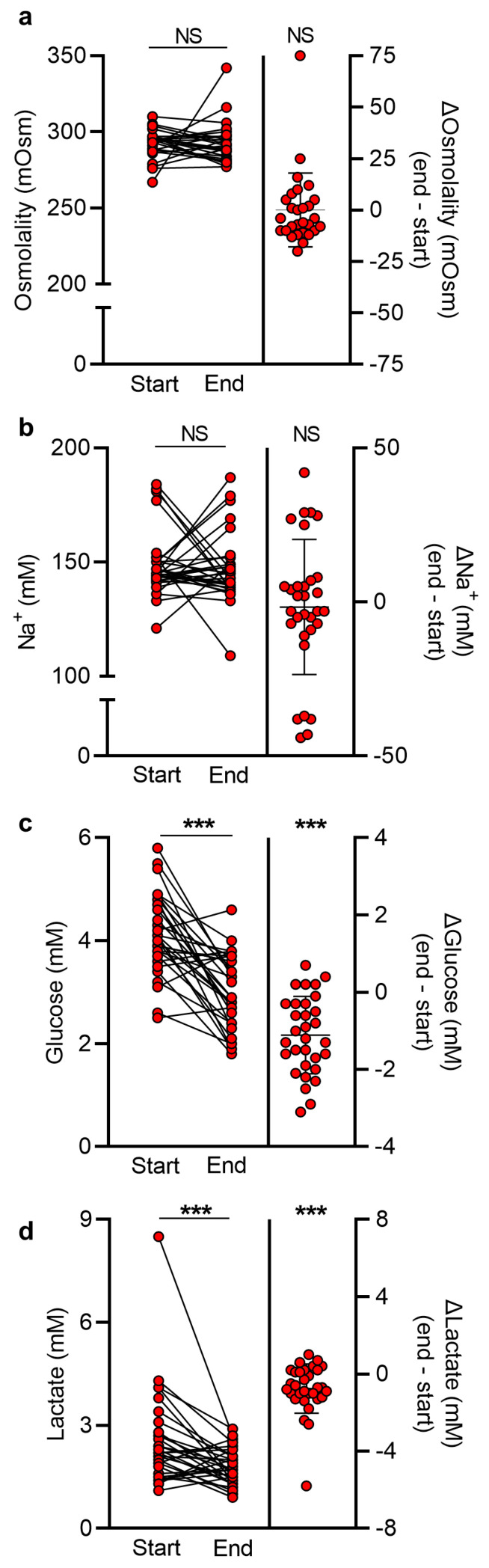
**CSF glucose and lactate disturbances in patients with SAH decline over time.** (**a**) CSF osmolality across the SAH patient group separated into start samples collected upon hospitalization and end samples collected prior to EVD removal (*n* = 28 of each, left panel). Average CSF osmolality change is depicted on the right (*n* = 28). (**b**) CSF Na^+^ concentration across the SAH patient group separated into start samples and end samples (*n* = 31 of each, left panel). Average CSF Na^+^ change (mM) is depicted on the right (*n* = 31). (**c**) CSF glucose levels across the SAH patient group separated into start samples and end samples (*n* = 31 of each, left panel). Average CSF glucose change (mM) is depicted on the right (*n* = 31). (**d**) CSF lactate levels across the SAH patient group separated into start samples and end samples (*n* = 31 of each, left panel). Average CSF lactate change (mM) is depicted on the right (*n* = 31). Statistical significance was tested with a paired two-tailed *t*-test or a Wilcoxon matched-pairs signed rank test, depending on normality. For determination of average changes, a one-sample two-tailed *t*-test or a one-sample Wilcoxon signed rank test was conducted and the results are presented as mean ± standard deviation. *** *p* < 0.001. NS = not significant.

## Data Availability

Anonymized data are available upon reasonable request to the corresponding author.

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
