# Peer review of "Posthemorrhagic Hydrocephalus in Patients with Subarachnoid Hemorrhage Occurs Independently of CSF Osmolality"

_ijms, 2023, doi:10.3390/ijms241411476_

Round 1

Reviewer 1 Report

This is a well-written, well thought-out and interesting study investigating the relationship between posthemorrhagic hydrocephalus and CSF osmolality. It addresses important research questions as osmotic effects are often neglected when discussing fluid dynamic disturbances of the CSF. I read it with interest and I do have a few questions for the authors:

1)    I think it is a bit unclear if all patients included in the study were confirmed as hydrocephalic, or if it was just a subset. I saw no data that validated the diagnosis of hydrocephalus in the manuscript, e.g., CSF volume, pressure, Evans’ index etc.

2)    Did the author’s test if the variables investigated (e.g., osmolality, Na+ etc.) varied/correlated with age? I also wonder if there were any relations between these variables and the time of CSF sampling, as well as the time between the first and second CSF sample? (since this time interval varied quite a lot within the group investigated)

3)    Figure 3 b seems to have a few values that deviate in different directions. Were there any common traits found in these individuals that could explain these groupings?

4)    The authors mention the different sampling sites (the ventricles vs. basal cisterns) as a limitation, and I agree. Are there any evidence in the literature that the authors are aware of that support these regions being similar/different in their content?

Minor comments:

Lines 70-74: At this point in the manuscript it is not made entirely clear what the current study does that reference 31 does not, and why the current study is performed as it is. Maybe this could be made clearer here?

Line 143: Why could they not be quantified? Was the reason the same for all of them?

Line 314: Maybe this could be rephrased. And maybe it could be stated again in the discussion that references 32-33 sampled CSF from the lumbar region.

Line 327: Yes, the small sample size is a major limitation of the current study, especially for the weaned and shunt group comparisons. But the results likely indicate that the difference (if existing) is not extreme. Regarding differences in osmolality, do the authors have any opinion of what effect size would be clinically relevant?

Other comments:

Definition of IVH is missing.

Some of the figures were blurry.

Grammatical comments:

Line 108: on average instead of in average?

Line 206: 3.3 should be 3.4.

Lines 262-264: I have a hard time reading this paragraph. Should it not be “…thereby do not contribute…” ?

Line 283: …observed in human…?

Reviewer 2 Report

This is a nice paper illustrating the important role of choroid plexus membrane transporters to regulate the biochemical composition of the CSF following subarachnoid hemorrhage.

The paper could be improved by adding some information about the neurological status of the study population, i.e., the WFNS and Fisher's scale sores of the cases.

Also, authors should discuss the role of impaired CSF absorption and its possible treatment with fibrinolytics (see references below). The statements that the pathophysiology of posthemorrhagic treatment is not known and that pharmaceutical treatment is not possible  should be revised.

1: Brinker T, Seifert V, Stolke D. Acute changes in the dynamics of the

cerebrospinal fluid system during experimental subarachnoid hemorrhage.

Neurosurgery. 1990 Sep;27(3):369-72. doi: 10.1097/00006123-199009000-00005.

PMID: 2234329.

2: Brinker T, Seifert V, Stolke D. Effect of intrathecal fibrinolysis on

cerebrospinal fluid absorption after experimental subarachnoid hemorrhage. J

Neurosurg. 1991 May;74(5):789-93. doi: 10.3171/jns.1991.74.5.0789. PMID:

1901600.

3: Germans MR, Dronkers WJ, Baharoglu MI, Post R, Verbaan D, Rinkel GJ, Roos YB.

Antifibrinolytic therapy for aneurysmal subarachnoid haemorrhage. Cochrane

Database Syst Rev. 2022 Nov 9;11(11):CD001245. doi:

10.1002/14651858.CD001245.pub3. PMID: 36350005; PMCID: PMC9644641.

4: Park YS. Fibrinolytic (Thrombolytic) Therapy for Post Intraventricular

Hemorrhagic Hydrocephalus in Preterm Infants. J Korean Neurosurg Soc. 2023

May;66(3):263-273. doi: 10.3340/jkns.2022.0258. Epub 2023 Jan 16. PMID:

36642948; PMCID: PMC10183265.
